# Testosterone and Androgen Receptor in Cancers with Significant Sex Dimorphism in Incidence Rates and Survival

**DOI:** 10.3390/cancers17213414

**Published:** 2025-10-23

**Authors:** Jianjian Lin, Jingwen Zhu, Jay Fowke, Ramesh Narayanan, Feng Liu-Smith

**Affiliations:** 1Tripill Biotechnolgy, Corp., Chapel Hill, NC 27516, USA; jianlin@unc.edu; 2Department of Pharmacy and Pharmaceutical Sciences, St. Jude Children’s Research Hospital, Memphis, TN 38105, USA; jingwen.zhu@stjude.org; 3Department of Preventive Medicine, College of Medicine, University of Tennessee Health Science Center, Memphis, TN 38105, USA; 4Department of Medicine, College of Medicine, University of Tennessee Health Science Center, Memphis, TN 38105, USA; 5Department of Dermatology, College of Medicine, University of Tennessee Health Science Center, Memphis, TN 38105, USA

**Keywords:** testosterone, androgen receptor, cancer, sex dimorphism

## Abstract

**Simple Summary:**

Sex differences play an important role in how certain cancers develop and progress. This study reviewed patterns of cancer incidence and survival between men and women, focusing on tumors of the esophagus, bladder, head and neck, lung, liver, kidney, stomach, and skin melanoma. We explored how male sex hormones (mainly testosterone) and the androgen receptor (AR) may influence cancer risk and outcomes. In some cancers, higher AR activity is linked to worse survival, suggesting that therapies targeting AR could help certain patients. However, the relationship between hormones and cancer is complex and not fully understood. More research is needed to clarify these effects and to develop treatments that consider biological sex as a factor in cancer care.

**Abstract:**

Several major cancer types exhibit significant sex dimorphism in incidence and survival. Whether and how sex as a biological factor impacts tumorigenesis, progression, and survival warrants full investigation, as such knowledge may lead to novel, precise prevention and treatment strategies. We reviewed epidemiological and molecular data on sex differences in cancers of the esophagus, bladder, head and neck, lung, liver, kidney, stomach, and skin melanoma, as well as the potential role of androgens and androgen receptor (AR) activity in these cancers. The potential molecular mechanisms are briefly discussed. Elevated testosterone (T) levels seemed to be associated with increased liver cancer and cutaneous melanoma incidences, and with reduced esophageal cancer risk. AR activity does not always correlate with T levels in tumorigenesis and progression. Higher AR expressions are associated with poorer survival in ESCC, whereas the role of AR in the survival of HNSCC and melanoma patients is inconsistent. The molecular impact of AR in liver cancer, kidney cancer, melanoma, and lung cancer is controversial. However, AR is likely to promote tumor growth and/or progression in esophagus, bladder, head and neck, and stomach cancers, and thus is associated with poor survival. Patients diagnosed with a tumor in this latter group could potentially benefit from therapeutic approaches targeting AR. Overall, the research on sex hormone androgens and AR in these cancers is limited. Further research is needed to determine a possible U-shaped relationship of T with cancer risk, and to decipher the role of testosterone and AR in some of these tumors to facilitate our understanding of sex dimorphism and to explore novel T/AR-based treatment options.

## 1. Introduction

Sex disparities in cancer incidence and mortality are well-documented across various cancer types, geographic regions, age groups, and time periods. While sex hormones like androgens, estrogens, and progesterone, along with their cellular receptors, are known to causatively impact cancers of the reproductive system (mainly, breast cancer, ovarian cancer, cervical cancer, and prostate cancer), their influence on cancers originating in non-reproductive organs remains less clear [1]. Emerging research suggests that sex hormones may impact the development and progression of non-reproductive cancers through various mechanisms, including dual roles as an oncogene and as a tumor suppressor, depending on cellular environment and cell lineage [2]. For example, an AR-modulated immune response can be either pro-inflammatory or anti-inflammatory [3]. These results were demonstrated in prostate cancer, where the roles of AR were more clearly understood. Similar results were also reported in non-reproductive cancers, with less clarity [1,4], and the goal of this review is to gain a comprehensive understanding of current discoveries of testosterone/AR in a few selected cancer types.

Specifically, we summarized epidemiological data and molecular evidence from the literature, taking into consideration sex differences in cancers arising from the non-reproductive system. It is noticeable that lifelong fluctuations in sex hormone levels may also be associated with the observed sex differences in cancer incidence and mortality. Our focus will be on androgen and the androgen receptor [5].

Recently, it was demonstrated that the binding patterns of AR in normal prostate cells on the target genes were significantly different as compared to tumor cells, highlighting a tumor suppressor role of AR in normal tissues, in contrast to a tumor-promoting role in cancer cells [2,6]. Correspondingly, both high and low levels of serum testosterone are linked to prostate cancer risk or worse prognosis [7]. Therefore, a U-shaped relationship was proposed to explain the complex relationship of this key sex hormone with prostate cancer [8]. In this review, we adopt this hypothesis and propose that the T/AR axis contributes to sex disparities in non-reproductive cancers not linearly, but in a context- and cancer-type-dependent manner, potentially also following a U-shaped model (Figure 1).

## 2. Overview of Androgens, Androgen Receptors, and Their Roles in Disease Development

### 2.1. Androgens and the Androgen Receptor

Androgens, also known as male sex hormones, are referred to a group of steroid hormones that play crucial roles in the development and maintenance of male characteristics and reproductive activity [9]. Androgens include androstenedione (A4), dehydroepiandrosterone (DHEA), DHEA sulfate (DHEA-S), testosterone (T), and dihydrotestosterone (DHT). They are synthesized from cholesterol, mainly in the testes, ovaries, and adrenal glands [9]. While present in both males and females, males typically have higher levels of androgens [10]. T is the primary androgen present in the circulation of mature mammals [11]. Sex hormones, including T and DHT, normally bind to sex hormone binding globulin (SHBG) in the circulatory system, and the free hormones are considered to be biologically active [12].

Androgens primarily exert their effects by binding to androgen receptors (ARs), encoded by the AR gene located on Chromosome X at Xq11-Xq12 locus [13]. This ligand-dependent transcription factor belongs to a nuclear hormone receptor (NHR) superfamily classified by protein homology, which includes other Type I class nuclear hormone transcription factors, such as estrogen receptors (ERs), glucocorticoid receptors (GRs), progesterone receptors (PRs), and mineralocorticoid receptors (MRs) [14]. AR is also known by the new nomenclature as NR3C4 (nuclear receptor subfamily 3, group C, gene 4). AR and other NHRs have three primary functional domains, including a N-terminal domain (NTD), a DNA-binding domain (DBD), and a ligand-binding domain (LBD) at the C-terminus, which has been extensively reviewed elsewhere [15,16].

### 2.2. Androgens Exhibit Both Receptor-Dependent and -Independent Effects

Inactive forms of AR are typically found in the cytoplasm, bound to heat shock proteins (HSPs); such an association ensures proper folding of AR [17,18]. Upon entry of androgens (such as T) into target cells by passive diffusion and binding to AR LBD, the AR undergoes a conformational change, leading to its dissociation with HSPs, homodimerization, translocation into the nucleus, and activation of AR-target genes [19] (Figure 2). Downstream signaling cascades are activated through two different mechanisms: the canonical (or genomic) pathway (Figure 2) and the non-genomic pathway [20].

In the current model for canonical AR signaling pathway, dissociation from HSP triggers the homodimerization of ligand-bound AR and its translocation into the nucleus, where dimerized AR interacts with androgen response elements (AREs) of target genes for transcriptional regulation. The modulation of gene expression governs processes such as development, homeostasis, and function of androgen-responsive tissues [14,21]. AR targets include genes regulating a range of cellular functions such as cell cycle progression (*CCND1*, *CDK4*, and *CDC25A*), apoptosis (*BCL2* and *MCL1*), DNA repair and genome stability (*Ku70/Ku80*, *BRCA1*, and *RAD51*), immune modulation (*IL-6* and *PD-L1*), cell invasion and tumor metastasis (*MMP9*, *VEGF*, and *EPHB2)*, metabolism *(HK2*, *LDHA*, and *ACLY)* and miRNAs.

While non-genomic (rapid and extranuclear) actions of androgens have been proposed, such as direct interactions with cytoplasmic targets like Src kinase or PI3K/AKT, leading to quick cellular responses [22]. These are controversial and less relevant to the chronic processes underlying cancer disparities discussed in this review. Non-genomic signaling is typically transient and observed mainly in in vitro models at pharmacological (supra-physiological) concentrations of steroids, with limited in vivo evidence supporting its role in sustained tumorigenesis or progression in non-reproductive cancers. Moreover, many reported non-genomic effects may be artifacts of high-dose experiments, confounded by overlapping genomic actions, or non-physiological responses rather than true endogenous mechanisms [23]. Thus, this review focuses on the well-established genomic pathway, as it better aligns with the epidemiological and molecular data linking T/AR to sex disparities in cancer incidence and survival.

### 2.3. Role of Androgen Receptor and Testosterone in Health and Development

In health, AR mediates T and DHT effects for male differentiation and maintenance, with fetal T-AR signaling forming genitalia [24] and pubertal AR-T promoting muscle, bone, and spermatogenesis [25]. Females rely on low T-AR for ovarian and metabolic balance [25], while AR supports non-reproductive anabolism and cognition across sexes [26], potentially influencing tumor dynamics in non-reproductive sites.

Mutations disrupt AR function, causing developmental issues like Androgen Insensitivity Syndrome from loss of function, leading to virilization failure and gonadal cancer risk [27], or spinal and bulbar muscular atrophy via CAG expansions, inducing neurodegeneration [28]. Somatic mutations drive prostate cancer progression [29], with germline variants linking to cancer susceptibility [30], highlighting mutational roles in disparities. Such mutations may exacerbate male-predominant cancers by altering AR sensitivity in non-reproductive tissues.

Utilizing the expression data from the GTEx project [31], we summarized AR expression in various organs/tissues (Figure 3A). The highest mRNA expression levels are found mostly in reproductive tissues such as the prostate, breast, ovary, vagina, and cervix, while the lowest expression is in brain tissues. All other tissues have various expressions. For most tissues, there seem to be comparable levels between males and females. Liver tissue unexpectedly has a level comparable to breast mammary tissues. For skin, suprapubic skin seems to have slightly higher expression than the sun-exposed lower leg tissues. For the esophagus mucosa, AR mRNA expression is low, consistent with previous IHC staining of Barrett Esophagus mucosa tissues, which showed negative staining for AR [32].

For a comparison between normal tissue and tumor tissue, we plotted the mRNA expression of AR using an online tool called TNMplot (https://tnmplot.com/analysis/ (accessed on 5 August 2025)) [33] (Figure 3B). Consistent with GTEx data, normal liver tissue and liver cancer showed relatively high AR expression, together with renal pelvis adenocarcinoma (Figure 3B). Skin data in this figure should be interpreted with caution because the tumor data are from melanoma, while the normal skin only contains a small portion of melanocytes.

### 2.4. Current Knowledge About Androgens and AR in Diseases

Both males and females require a balance between androgens with estrogens in the system to maintain fertility and reproductive organ function. For example, polycystic ovary syndrome (PCOS), leading to irregular menstruation and difficulty in becoming pregnant, is found in approximately 4 to 20 out of every 100 women of reproductive age. The underlying causes for PCOS may vary; PCOS is commonly characterized by elevated androgen levels produced by the ovaries to block ovulation [34]. Similarly, 20–30% of male infertility cases were found to be associated with low T levels or increased levels of luteinizing hormone (LH), leading to a decrease in sperm count [35].

An androgen imbalance may also play a crucial role in the development of cancers in reproductive organs, such as prostate cancer and androgen-sensitive breast cancer [36,37]. Notably, the role of androgens and AR signaling pathways in prostate cancer, the most frequently diagnosed cancer in American men and the second leading cause of cancer-related deaths, is the most studied [38].

Besides their roles in the reproductive system, androgens are also important for the development and maintenance of many non-reproductive physiological tissues, such as muscle mass and bone density [9,39]. Moreover, several types of cancers originated in non-reproductive tissues, such as lung cancer, bladder cancer, melanoma, and glioblastoma, have been associated with the activation of androgens/AR signaling pathways for proliferation or enhancing survival of cancer stem cells [40,41]. Although preclinical data suggested blockage of AR signaling is promising to inhibit proliferation of these tumor cells, less is known about the exact mechanism, and thus it remains a challenge in blocking androgens/AR signaling pathways in therapeutic strategies against non-reproductive cancers. For instance, a key consideration when selecting an AR antagonist for glioblastoma treatment is the ability of the drug to cross the blood–brain barrier [42]. Additionally, determining the heterogeneity of AR isoform expression is crucial for appropriately applying AR inhibition in these cancers to ensure targeted and effective treatment.

A clue to the significance of androgens in these cancers may come from a better understanding of sex differences in cancer risk. Since androgen levels and perhaps also AR levels differ between males and females [40], it is of great interest to determine whether androgens, particularly T, and AR are linked to sex-specific differences in the incidence and survival outcomes of cancers arising in non-reproductive tissues. In the next section of this review, we will outline the known involvement of T and AR in cancers originating from non-reproductive systems and attempt to understand their possible role in sex disparity.

### 2.5. Overview of AR in Cancer

As shown in Figure 3B, AR expression is enhanced in some cancer types and decreased in other types compared to their corresponding normal tissues, suggesting a complex role of AR, as well as tissue-specific functions of AR in cancers. AR plays a role in almost all aspects of cancer development, such as tumor cell proliferation, apoptosis, invasion, metastasis, tumor microenvironment modulation, tumor immune responses, drug resistance, and patient survival [43].

While the role of AR in initial tumorigenesis can be either oncogenic or tumor suppressive [44,45,46,47,48], in established cancer cells, AR generally stimulates cell proliferation and promotes tumor growth [49]. The oncogenic function of AR is primarily through its transcriptional activity and its association with other oncogenes such as EGFR, RAS family, and MAPK pathway, as well as cell type-specific co-regulators [49,50,51]. The tumor suppression function is partially through AR targets, such as PTEN and DNA damage response genes [52,53]. However, the same DNA damage response may also lead to drug resistance [49].

AR plays multifaceted roles in regulating cancer cell invasion and metastasis, with effects that vary across tumor types and microenvironments in cell invasion and tumor metastasis. For example, AR promotes metastasis in lung cancer by modulating the miR-23a-3p/EPHB2 signaling axis. AR was reported to increase hematogenous metastasis while suppressing lymphatic spread, mediated by differential regulation of VEGF-A and VEGF-C via miR-185-5p [54]. Mechanistically, AR can act through transcriptional regulation of metastasis-related genes and microRNAs, influencing epithelial–mesenchymal transition and angiogenesis. Therapeutically, targeting AR signaling may offer site-specific benefits, especially when combined with other clinical drugs.

The relationship between circulating androgen levels and tumor-intrinsic AR activity is quite complex. Circulating testosterone declines with age, obesity, chronic diseases, and chemotherapy [55,56,57]. Overall, hypogonadism is associated with poorer cancer survival [58,59]. Higher circulating testosterone levels could directly activate tumor AR activities, but tumors can also develop ligand-independent sustained AR activities through mechanisms such as AR mutation, gene amplification, or even intratumoral androgen synthesis [60]. In prostate cancer, both high and low serum total testosterone levels have been associated with increased risk [8], suggesting a non-linear relationship. However, blocking tumor AR activity is a standard therapeutic method. AR blockade has now been tested in a number of other cancer types, with various results [61]. Similar observations in other contexts support this complexity; for example, Di Donato et al. reported that ligand-activated AR drove melanoma invasiveness independent of systemic androgen levels [62].

## 3. An Overview of Sex Disparities in Cancers in Non-Reproductive Organs

Male predominance is found in multiple cancers in non-reproductive systems, including esophageal cancer, bladder cancer, head and neck cancer, liver cancer, oral larynx cancer, kidney cancer, and stomach cancer (Table 1). Using the data from the Surveillance, Epidemiology, and End Results (SEER) Program of the National Cancer Institute (downloaded by SEERStat9.0.41 software), the age-standardized incidence rates (ASRs) in men and women from 2000 to 2022 are plotted in Figure 4. Esophageal (SEER site code C15) and bladder cancers (C67) exhibited the greatest sex differences, more than fourfold of ASR in males as compared to females (M/F ratio), followed by head and neck cancer (C00–C14 and C32, with an M/F ratio of 2.84), liver cancer (C22), kidney cancer (C64.9), stomach cancer (C16), and cutaneous melanoma (C44). Major cancer types with an M/F ratio > 1.5 are included in this review, as well as highly prevalent lung cancer, which had an M/F ratio of 1.37.

### 3.1. Esophageal Cancer

Esophageal cancer incidence is 6.3 per 100,000 person-years globally and exhibits a substantial sex difference [63,64] (Table 1). Its 5-year relative survival rate is only 15–20% [65]. The incidence and histological types of esophageal cancer vary by region. Esophageal squamous cell carcinoma (ESCC) is more prevalent in developing countries, while increasing cases of esophageal adenocarcinoma (EAC) have been observed in recent years in developed nations, including the United States and Western Europe [63,64].

It is estimated that 22,070 new cases of esophageal cancer will be diagnosed in the United States in 2025, and about 80% of them will be in males [66]. This trend is also revealed by data obtained through SEER from 2000 to 2022, with the incidence rates for esophageal cancer of 7.4 per 100,000 for males and 1.8 per 100,000 for females (Figure 4, Table 1), resulting in a male-to-female incidence rate ratio of approximately 4.16 (Table 1). The odds for EAC and ESCC are estimated to be 7–10 and 3–4 times higher in males than in females, respectively [64]. The established risk factors, like obesity, gastroesophageal reflux disease (GERD), or smoking, cannot fully explain the male predominance, implicating androgens as a contributing factor [67,68].

Whether sex hormone levels are associated with esophageal cancer remains inconclusive. One study showed that higher concentrations of DHEA but not T or DHT were associated with reduced risk of EAC [69]. A separate nested case–control study showed that higher T levels or higher T/E2 (testosterone/estradiol) ratios were associated with a decreased risk of EAC [70] (Table 2). A meta-study combined these two studies validated the association of T with EAC [70], which was not confirmed in a later meta-study with five studies [71] (Table 2). A potential confounder is perhaps BMI, as it is known that testosterone levels decrease with increased BMI, and after adjusting for BMI, the association of T with EAC is no longer significant (Table 2).

AR showed varied expression levels in ESCC cell lines and tumors from patients [72], and higher expression was found in invasive ESCC tumors [73] (Table 3). Consequently, high tumor AR levels were associated with poor patient survival, especially for smoker patients, consistent with the finding that AR promoted tumor cell proliferation as well as invasion [72]. A second study validated the association of high tumor AR levels with poor ESCC patient survival [74].

The mechanism through which AR acts in ESCC is not fully understood. Existing evidence shows that AR may exert its effects by upregulating matrix metalloproteinase 2 (MMP2) and activating the AKT signaling pathway [73], or through a feedback loop via pro-inflammatory cytokine interleukin-6 (IL-6), concomitant with activation of the STAT3 oncogenic signaling [72]. A ChIP-seq analysis indicated that GATA3 was a key co-regulator of AR in ESCC, and DUSP4 and FOSB were the key negatively regulated targets of AR [74]. Lower expression of DUSP4 or FOSB seemed to be associated with worse disease-free survival in patients [74]. A more recent study of this same ChIP-Seq experiment identified an additional AR target gene in ESCC: UGT2B15 (uridine diphosphate glucuronosyltransferase family 2 member B15), with AP-1 as an important co-regulator [75]. Inhibiting both AP-1 and AR exhibited strong effects in inhibiting cell invasion [75]. The AR targets and co-regulators are listed in Table 3.

Although cellular T or DHT is normally believed to upregulate AR levels, whether this regulation occurs in esophageal cancers remains uncertain. In ESCC and EAC, lower circulating T levels have been associated with increased cancer risk, whereas higher AR expression is observed in more advanced tumors and correlates with poorer survival [73]. An extensive investigation is warranted into the roles of T, DHT, and AR function in esophageal cancers.

### 3.2. Bladder Cancer

Bladder cancer is one of the top non-reproductive cancers showing a male-predominant pattern [76]. As shown in Table 1, the incidence rates for bladder cancer were 34.5 per 100,000 for males and 8.4 per 100,000 for females, indicating a 4.1 times higher incidence rate in men as compared to women. Following this trend, there will be 65,080 new cases in men and 19,790 new cases in women diagnosed with bladder cancer in 2025 [66]. A recent meta-analysis based on nine previous studies showed that nulliparous women and women with early menopause exhibited a significantly higher risk of bladder cancer [136], suggesting a link with sex hormones [137]. Conversely, women are more likely to be diagnosed with advanced stages of the disease [138,139]. Furthermore, female sex is independently associated with disease recurrence and worse cancer-specific survival [140,141].

The poorer outcomes for women in bladder cancer were partly attributable to diagnostic delays, as hematuria in women is frequently misattributed to urinary tract infections, resulting in later-stage diagnoses [142]. Moreover, women are less likely to receive guideline-concordant treatments such as neoadjuvant chemotherapy and radical cystectomy, even when matched for disease stage and comorbidities [142]. The survival gap is most pronounced in advanced-stage disease, where female patients demonstrate significantly lower overall survival than their male counterparts. The underlying biology and genomics mechanism may be related to sex hormone signaling, but this requires further investigation [143].

Tobacco usage is the strongest risk factor for bladder cancer, accounting for 50–65% of all cases [144]. Smoking was assumed to be a major cause of sex differences in bladder cancer, as there were more male smokers in general. However, when smoking intensity was stratified, men consistently showed higher incidence rates than women in each stratum, suggesting that smoking alone is insufficient to explain the sex disparity [145]. Zhu and Zhao summarized various factors that may impact the sex differences in bladder cancer, including biological differences and occupational differences [139]. Together with the above-mentioned survival disadvantage in women, it is vitally important to understand the mechanism behind the sex difference in bladder cancer incidence and survival.

Epidemiological research has investigated the association between androgens and bladder cancer. A recent case–control study in China, involving 147 male bladder cancer patients and 154 healthy controls, reported that cancer patients exhibited higher serum total T levels [146]. However, a study using the large UK Biobank data did not find such an association, either in men or post-menopausal women [111]. A two-sample Mendelian randomization (MR) study also found no relationship between genetically predicted T levels and bladder cancer risk in men [147] (Table 4).

Higher AR levels were found in bladder tumors compared to normal bladder mucosa, and they were higher in males compared to females [77]. The potential role of AR in bladder cancer development was described as early as 1975. In this mouse study, male mice showed a higher incidence of chemically induced bladder tumors compared to female mice [78]. Bladder tumors were absent in AR knockout mice, which underscored the pivotal role of AR in carcinogen-induced bladder carcinogenesis [78]. A mouse model with urothelial-specific AR knockout showed that DNA damage was reduced and TP53 was increased in AR-negative cells as compared to wild-type urothelial cells [46]. Another experiment with AR knockdown showed reduced levels of CD24, a key driver for metastasis [79]. Supportive of this finding, in a cohort of 10,720 male patients with bladder cancer, patients who received 5α-reductase inhibitors, which decrease the AR activity by reducing DHT levels, exhibited improved disease-specific survival [80]. Furthermore, androgen depletion in AR-positive human bladder cancer cells in vitro, or castration in mice, inhibited tumor cell growth, as did the AR knockdown [81,82]. AR expression was detected in only a subset of bladder cancer cell lines and bladder tumors [77,83]. Some bladder tumors contained little androgen, with nearly undetectable DHT and castration-level of T [84]. Consistent with a tumor suppressor role of androgens and/or AR in bladder cancer, the absence of AR expression was associated with recurrence in non-muscle-invasive bladder cancer [85,86]. Still, a meta-analysis revealed that AR levels were positively correlated with the tumor grade of bladder cancer, but not with susceptibility or tumor stage [87]. Whether the possible tumor suppressor role of AR and/or androgen explains the disadvantage of female survival in bladder cancer warrants further investigation.

In terms of molecular mechanisms, multiple downstream targets or signaling cascades of AR may be involved, such as EGFR, AKT, matrix metalloproteinases, UDP-glucuronosyltransferases (UGTs), and β-catenin/WNT pathway [88,89,90], which were extensively reviewed by Li et al. [77]. A recent study suggested that AR-regulated circular RNA activated KRAS signaling, thereby enhancing bladder cancer invasion and chemoresistance [91]. Of particular interest regarding sex differences is the GATA3 gene, a suggested prognostic biomarker for bladder cancer [92,93]. GATA3 expression is significantly higher in bladder tissues of female mice compared to male mice, and its expression was increased by orchiectomy in males but decreased by ovariectomy in females [94]. Loss of GATA3 in a subset of bladder cancer, especially in high-grade tumors, was correlated with higher AR activity in males [94].

### 3.3. Head and Neck Cancer

Head and neck cancer ranks as the seventh most common cause of cancer-induced mortality, encompassing a group of malignancies from the anatomical sites of the upper aerodigestive tract, the paranasal sinuses, and the salivary glands [101], with major subtypes including head and neck squamous cell carcinoma (HNSCC), oral squamous cell carcinoma (OSCC), oropharyngeal squamous cell carcinoma (OPSCC), and so on. HNSCC, the most prevalent subtype (accounting for over 90% of all cases) [102], has a five-year survival rate, less than 50% in advanced stages, and males have up to a fivefold higher risk in all ethnic groups when compared to females [103,104]. The SEER data revealed incidence rates for HNC of 23.0 per 100,000 for males and 8.1 per 100,000 for females, resulting in an M/F ratio of approximately 2.8 (Figure 4, Table 1). In 2025, it is estimated that approximately 59,660 new cases of cancer in the oral cavity and pharynx will be diagnosed in the United States, and 42,500 of them will be men, which is two to three times higher than that for women [66]. The major risk factors for head and neck cancer include tobacco smoking, chronic alcohol consumption, and infection with human papillomavirus [105].

To date, large-scale prospective studies have not demonstrated a clear association between T levels and the overall incidence of head and neck cancers (HNC). For instance, one study that examined sex hormone levels across multiple cancer types found no significant link between T and cancers of the oral cavity, pharynx, or larynx [153]. However, hormonal factors may not be ignored because a pooled analysis of 11 case–control studies indicated that women who had used hormone replacement therapy (HRT) experienced approximately a 42% reduction in the odds of developing HNC (OR ≈ 0.58) compared to those who had never used HRT [154].

A possible interplay between HPV positivity and AR signaling has been suggested in HNSCC. Mohamed et al. (2018) observed that HPV-positive OPSCC (oropharyngeal squamous cell carcinoma) tumors were associated with increased AR expression as compared to HPV-negative tumors [155], and oropharyngeal tumors were more likely to express sex hormone receptors, including AR and ERa, as compared to other sites in HNSCC [155,156]. However, the biology of OPSCC may differ from the rest of HNSCC.

The significant sex difference in HNSCC has prompted hypotheses of a synergistic interplay among smoking, alcohol consumption, and sex hormones [157]. Tobacco synergizes with AR, as smoke components upregulate AR expression and activity, boosting HNSCC cell proliferation via EGFR/AKT [158], explaining male bias from higher smoking rates [157]. Alcohol shows limited direct AR synergy, often suppressing T via metabolism or estrogen conversion [159], but may indirectly enhance AR signaling through Golgi fragmentation in models [160], potentially cooperating with inflammation in HNC, though evidence is sparse [161]. Future studies should clarify AR–alcohol crosstalk in HNC cohorts.

AR expression varies in HNSCC at different sites; for example, the AR mRNA level was upregulated in laryngeal squamous cell carcinoma [100,162] (Table 3). Upregulation of AR and a greater distribution into cytoplasm (≥20%) were associated with poor progression in OSCC [97], consistent with in vitro studies, which demonstrated that AR suppression reduced tumor cell growth by inducing apoptosis [157]. Consistent with these results, an immunohistochemical study showed that when more than 20% of OSCC tumor cells showed cytoplasmic AR staining, epithelial Ki67, lymphocyte VEGF, and macrophage MMP9 were all significantly positively associated with AR levels, while epithelial HIF1b and macrophage VEGF were negatively associated with AR levels in metastatic tumors [98]. In AR+ OSCC cell line SCC9, DHT induced AR expression and activation, which led to enhanced cell migration and aggressiveness by induction of EGFR and AKT [50]. This signal pathway and effect were absent in AR-cell lines, underscoring the direct involvement of the AR gene [50]. AR also cooperated with co-activator RBAP48 (Retinoblastoma-associated protein 48), a protein highly expressed in OSCC, to activate AR targets. Both AR and RABP48 at higher levels were associated with worse survival of patients [99]. AR co-regulators and targets are summarized in Table 3. Hence, AR was suggested to be a prognostic biomarker for OSCC and a promising target for therapeutic intervention in OSCC.

Salivary gland cancers (SGCs) are relatively rare and usually diagnosed in older people (>60 years) [163,164,165]. Among SGCs, salivary duct carcinoma (SDC) is a highly aggressive subtype, with AR positivity observed in around 75–89% of cases [96,166,167]. Hence, it was not a surprise that there were case reports showing that SDCs were responsive to androgen-deprivation therapy [168,169]. However, a Phase II clinical trial with Enzalutamide in AR+ SGC patients failed to meet the endpoint expectations [170]. The exact mechanism of AR function in SGC is largely unknown. It was reported that AR overexpression in SGC might be through the increased copy number of chromosome X or TP53 mutation [171]. Genomic analysis revealed that AR might interact with transcription factor FOXA1 (Forkhead box protein A1) to activate the downstream target genes [166,171,172].

### 3.4. Liver Cancer

Liver cancer, primarily hepatocellular carcinoma (HCC), is a significant global health concern, ranking as the third leading death-causing cancer in 2022 [107]. It was reported to have 2–4 M/F ratios in the age-standardized incidence rates across all regions in the world [173,174]. In the SEER data, the incidence rate for liver cancer was 12.8 per 100,000 for males and 4.7 per 100,000 for females, with an M/F ratio of 2.8 (Table 1). Despite this constant observation, the reason underlying the sex disparity remains unclear. Risk factors for liver cancer include viral hepatitis, aflatoxin exposure, alcohol consumption, diabetes, and obesity. However, it is hard to explain the consistency of sex disparity since the prevalence of these factors is changing over time in different populations [175]. This led to the hypothesis that the biological differences between males and females, such as sex hormones, may also play a critical role in liver cancer.

Multiple prospective studies have reported positive associations between T and liver cancer incidence. The UK Biobank study showed that total T and SHBG, but not free T, were significantly associated with liver cancer in men, but not in women [111]. A meta-analysis of 29 prospective studies found that higher T levels were associated with elevated risk of liver cancer, with a stronger effect in men [71]. An earlier prospective cohort study observed that men with higher total T had an increased risk of early mortality after cancer diagnosis, while women showed the same effect only in the highest quintile [176]. In an early diethylnitrosamine (DEN)-induced liver cancer mouse model, castration reduced both tumor incidence and tumor size in male mice [177]. Taking it together, current studies suggest that elevated T levels may be linked to an increased risk of liver cancer as well as mortality.

AR expression was found in both normal liver tissue and HCC, with an overall higher expression in tumor regions [178,179]. In one report, AR expression levels were positively correlated with the recurrence rate of HCC and negatively correlated with the survival rate, indicating AR as a promoter in liver cancer [108]. These results correspond with the above-mentioned results of T levels. Furthermore, within similar serum T levels, the absence of AR postponed or lessened the development of carcinogen-induced HCC [109]. However, loss of AR in mice bearing HCC led to more metastasis [110], corresponding to another report where a patient with higher AR expression in tumors showed better overall and disease-specific survival [180]. Taken together, it was suggested that AR played a dual role in liver cancer: a promoter in triggering tumorigenesis and a suppressor at advanced stages [110]. The controversial results regarding AR’s impact on survival outcome warrant further clarification [181].

The molecular studies suggested several instances of AR signaling in HCC (summarized in Table 3): upregulating EZH2 expression which correlates with tumor progression and poor prognosis [182]; promoting arachidonic acid metabolism and angiogenic tumor microenvironment in AFP (alpha-fetoprotein)-negative HCC [183]; and serving as a molecular docking target for metformin, resulting in lower AR at protein level and enhanced ferroptosis [180]. HBV (hepatitis B virus) increased AR expression, which in turn, upregulated BIRC7, IGFBP3, and NTSR1, leading to increased proliferation of HCC cells [184]. It was also suggested that AR (together with the estrogen receptor) interacts with HBV and HCV (hepatitis C virus) in a sex-specific manner, leading to sex-differentiated immune response, inflammatory damage, liver fibrosis, and carcinogenesis [185]. In addition, the polymorphism of AR (number of CAG repeats) affects liver cancer differently in males and females, which also likely contributes to sex disparities of HCC. Yu et al. reported that shorter AR alleles (i.e., fewer repeats) conferred a higher risk for HCC in men with HBV, whereas higher incidence rates were found in HBV female carriers with longer AR alleles [186].

### 3.5. Kidney Cancer

Since the 1970s, the global incidence of kidney cancer has been increasing [187]. In 2025, the National Cancer Institute estimated approximately 80,980 new cases in the United States, with 52,410 male cases [66]. Renal cell carcinoma (RCC) is a leading cause of cancer death, ranking sixth among men and eighth among women [188,189]. As shown in Figure 4 and Table 1, incidences of kidney cancer were 20.1 per 100,000 for males and 10.0 per 100,000 for females, resulting in an M/F ratio of 2.0. This male predominance is consistent across various age groups, time periods, and geographic locations [112,190,191]. The reasons behind these sex differences are multifaceted, involving both biological and environmental factors. Established modifiable risk factors, such as smoking and hypertension, contribute significantly to RCC incidence, and hormonal influences, particularly the role of androgens and the AR, have been implicated in RCC development and progression [113,192]. In particular, by upregulating 20-HETE production, which elevates renal vascular tone and blood pressure, androgen potentially creates a pro-tumorigenic environment in the kidney [114].

Epidemiological studies on the association between androgens and kidney cancer risk are limited. Two UK Biobank studies showed no significant association between baseline T (total or free) and risk of kidney cancer in either sex [111,193]. RCC is the predominant form of kidney cancer, with clear cell RCC (ccRCC) being the most common subtype [115,194,195]. AR expression was positively correlated with tumor-originated vasculogenesis in ccRCC patients [116], suggesting a tumor-promoting role, which was validated in in vitro experiments, possibly through modulating the lncRNA-TANAR/TWIST1 signaling pathway [116]. Consistently, AR suppressed the expression of a circular RNA named circHIAT1 by downregulating its host gene, resulting in enhanced ccRCC cell migration and invasion [117]. In contrast, other studies showed the opposite effect of AR in ccRCC: higher expression of AR correlated with better overall survival in both males and females [118,119,189]. Higher AR expression was also correlated with male patients, lower tumor grade, and early stages of RCC [120]. Again, the role of AR in ccRCC development and progression is inconclusive: even though AR was associated with tumor vasculogenesis and invasion, higher tumor AR levels were associated with better patient survival.

### 3.6. Stomach Cancer

Despite a decline in both incidence and mortality rates over recent decades, gastric cancer remains one of the most lethal malignancies [123]. In the United States, it was estimated that there would be 17,720 new stomach cancer cases in 2025, among which 12,580 would be males [66]. This male-dominant trend is also exhibited by the SEER data (2000 to 2022), which shows the incidence rates of 10.9 per 100,000 for males and 6.1 per 100,000 for females (Figure 4), indicating an M/F ratio of 1.80 (Table 1).

Current epidemiologic data do not suggest a clear association between T and stomach cancer risk. One systematic review of prospective cohorts of 11 studies concluded that circulating T was not significantly associated with gastric cancer risk in either sex [124]. The UK Biobank study also revealed no significant association of total or free T levels with stomach cancer, neither in men nor in women [111]. However, one study indicated that elevated SHBG was associated with increased gastric cancer incidence in men only [71]. This association was validated in the UK Biobank study, which was not found in post-menopausal women [111]. Hence, although T alone is not associated with stomach cancer, higher SHBG seems to confer a higher risk in men.

Research has indicated that disruptions in AR homeostasis might contribute to the higher prevalence of gastric cancer in men compared to women [125]. Further research has identified AR variants in gastric cancer tissues and cell lines, implicating them in promoting tumorigenesis and metastasis [126]. Additionally, recent research has discovered that AR activation leads to increased expression of LAMA4, which in turn induces resistance to cisplatin in gastric cancer cells, highlighting a potential mechanism by which AR contributes to treatment resistance [196]. In addition, AR was frequently overexpressed in gastric cancer tissues compared with normal gastric mucosa [121,197]. Hou et al. further explained that increased AR expression, facilitated by the co-factor YAP1, drove the development of chemoresistance in gastric cancer [198]. Therefore, AP appears to contribute to chemoresistance via multiple pathways. AR overexpression was also associated with nail, beta-catenin, Twist1, and STAT3, all of which led to worse outcomes [122].

### 3.7. Lung Cancer

Lung cancer remains the leading cause of cancer-related deaths worldwide, accounting for approximately 1.8 million fatalities annually and remaining the top death-causing cancer in the United States [199]. The major types of lung cancer include small cell lung carcinoma (SCLC), non-small cell lung cancer (NSCLC), the latter of which comprises lung adenocarcinoma (LUAD), lung squamous cell carcinoma (LUSC), and large cell carcinoma. The male predominance in lung cancer incidence is observed in the SEER data (2000 to 2022) with an M/F ratio of 1.37 (Table 1). However, the sex difference is diminishing and even reversed in recent years (Figure 4 and [135,200]). In the United States, it is estimated that there will be 226,250 lung cancer cases in 2025, with male cases of 110,680, less than the female cases (115,970) [66]. But women are often diagnosed with lung cancer at a younger age, in earlier stages, and generally have a more favorable prognosis and better 5-year survival rates compared to men, particularly in (NSCLC) [133].

The relationship between T and lung cancer remains inconclusive. One MR study found no relationship between T and lung cancer risk in either men or women [201], whereas another MR study showed that, based on 74 SNPs, higher levels of bioavailable T were associated with a lower risk of LUSC and SCLC but not LUAD in men [202]. No association was found in serum hormone levels of total T or free T or SHBG with lung cancer in men or post-menopausal women in a large UK Biobank study [111].

AR expression has been detected in both SCLC and NSCLC, including LUAD—the most prevalent type of NSCLC in non-smokers and young people [199,203,204]. However, the role of AR in lung cancer remains unclear. In A549 cells, an NSCLC cell line with relatively high AR expression, AR and EGFR (a crucial oncogene in lung cancer) collaboratively promote proliferation by activating p38 MAPK signaling [205,206]. AR also promoted invasion and metastasis via the miR23a-3p/EPHB2 pathway [134]. Treating A549 cells with luteolin reduced AR expression and subsequently inhibited cell proliferation [207]. Similarly, prostate cancer patients receiving androgen-deprivation therapy (ADT) showed lower secondary lung cancer incidence when compared to those in the non-ADT group [150].

Consistently, exposure to androgen pathway inhibition (primarily 5-alpha reductase inhibitors, which reduce DHT levels) led to greater survival of lung cancer patients regardless of age, stage, and histology [151]. On the contrary, other reports found that the presence or level of AR was positively correlated with better survival in LUAD patients [48,152]. This hypothesis was supported by the observation that AR inhibited tumor cell invasion in NSCLC via the circular-SLCO1B7/miR-139-5p axis and reduced oncogene D52 expression [208]. It is also likely through tumor-promoting miR-224-5p, which targets AR and inhibits AR expression [48]. Of particular note, given the distinct biology of LUAD, LUSC, and SCLC, we analyzed androgen-related findings by histologic subtype; associations observed in LUAD were not assumed to generalize to LUSC or SCLC.

### 3.8. Melanoma

The global incidence of melanoma has been rising in recent decades and is projected to continue increasing [209]. In 2025, it is estimated that 104,960 individuals in the United States will be diagnosed with invasive melanoma, with 60,550 male and 44,410 female cases [66]. Risk factors for melanoma include UV exposure (both from sunlight and artificial sources like tanning beds), fair skin, a history of sunburns, the number of moles, a family history of melanoma, and a weakened immune system [127]. The risk increases with age, with the average age of diagnosis being 65 [127]. Meanwhile, females exhibit higher incidence rates of melanoma compared to males at a younger age, whereas males have higher incidence rates at an older age [128]. Men have a lower melanoma survival rate compared to women after adjusting for age, stage of disease, and other prognostic factors [129].

The relationship between androgens and melanoma has not been extensively investigated, and results are mixed. Based on the observation of female survival advantage in melanoma, it was suggested in 1980 that melanoma might be an androgen-dependent tumor [130]. But later studies generally did not support this hypothesis due to a lack of apparent AR expression in most normal melanocytes or melanoma cells [53].

A recent study using UK Biobank data showed that higher total T and free T levels were associated with a higher risk of melanoma in men but not in post-menopausal women [111]. This study underscores the importance of investigating sex hormone impact on melanoma risk. However, this study did not fully adjust for sun exposure, which might confound the association between T levels and melanoma [111], as it has been shown that chronic UVB exposure increased T levels in male but not in female mice [131]. Men showed peak T levels in the summer or fall months [131,132,210]. Therefore, further clarification is needed. In another study, male pattern baldness, which is often associated with high T levels, was found to have a strong association with melanoma risk [211]. However, based on the MR analysis, the authors concluded that this association is likely driven by sun exposure rather than T expression [211]. Nevertheless, there is a potential crosstalk between T and sun exposure because a fish melanoma model showed that UVB irradiation was able to rapidly up-regulate AR expression [212], consistent with UV-increased T levels in humans.

Another MR study using instrumental T variants in the whole population without separating susceptibility by sex showed no causal association between T with melanoma risk [213]. This study failed to account for the markedly different genetic determinants of T between sexes [12], rendering the conclusion highly unreliable.

More recent molecular studies have shown AR expression in melanoma cell lines and tumors [214,215]. AR exhibits a seemingly controversial role in melanoma development and survival. Morvillo et al. demonstrated that androgens significantly stimulated proliferation in a human melanoma cell line expressing an atypical form of AR, while treatment with the androgen antagonist flutamide reversed these effects [216]. Min et al. also showed that AR suppression reduced melanoma tumor cells’ proliferation by increasing DNA damage and activating a STING-dependent inflammatory response [53]. Further, AR signaling was reported to drive melanoma invasiveness through tumorigenic fucosylation or the MITF/AXL axis [217,218]. Specifically, AR activated transcription of the FUT4 gene, which, in turn, fucosylated L1CAM at the adherence junctions, leading to increased metastasis [217]. AR also increased melanoma invasion through regulating the miRNA-539-3p/USP13/MITF/AXL signals. AR functioned through binding to AREs in the promoter region of the host gene for miR-539-3p and activated its expression [218]. In this study, AR+ melanoma patients were shown to have worse overall survival [218]. Another study did not find the association of baseline AR levels with patient outcomes [41], but they found that AR signaling was induced by BRAF/MEK inhibitors, and blocking this increase in AR improved patient responses to the targeted therapy [41]. These results, however, somewhat contrast with the epidemiological findings where patients in the TCGA dataset with higher AR levels showed better overall survival [40]. Previous publications also showed that lower T levels were associated with poorer cancer survival [219], particularly in hypogonadism patients [58,59]. A mouse study revealed that castration increased melanoma tumor burden, but supplementing T reversed it [220]. It was then suggested that extreme T levels were potentially harmful. Such non-linear observations were also reported in prostate cancer [8,221]. Based on these conflicting observations, we propose that a non-linear T/AR axis also exists in melanoma, as shown in Figure 1.

In melanoma, AR seemed to be heavily involved in the regulation of the tumor microenvironment and immune responses. A genomics study using the TCGA-SKCM data revealed that increased AR signal was significantly associated with exhaustion of CD8+ T cells, and blocking AR signaling could synergize with T cell-based cancer immunotherapy [4]. Inhibiting AR activities was shown to increase immune cell abundances and was associated with higher IFNγ pathway activity, leading to better response to ICI treatment [222].

## 4. Summary and Conclusions

The effect of androgens (including SHBG) is summarized in Table 4. No solid evidence shows an association of T levels with head and neck cancer, bladder cancer, kidney cancer, or lung cancer. Evidence may suggest a sex-specific link of T or SHBG with liver cancer, stomach cancer, and melanoma, with higher T associated with elevated risk. Expression levels of AR seem to tell a different story (Table 5): although there are controversies, inhibition of AR at the cellular level generally resulted in decreased cell proliferation. However, in liver cancer, kidney cancer, melanoma, and lung cancer, higher tumor AR levels are apparently associated with better patient survival.

Given all the above mixed results, we found a previous hypothesis regarding a U-shaped relationship of T with prostate cancer is interesting [8] (Figure 1). Specifically, Salonia et al. found that when serum total T was grouped into low, middle, and high levels, both low and high levels were associated with a higher risk of prostate cancer [8]. This may imply that appropriate levels of T are key in maintaining tissue homeostasis, including the non-reproductive tissues. Too high or too low levels of T may be associated with cancer risk. When only linear regression, or regression assuming a linear relationship, is used, the results can be mixed. While high levels of T apparently stimulate AR activities through ligand binding, evidence also showed that low levels of T increased AR protein stability in a compensatory manner [224]. Hence, it is likely that T/AR activities also follow a U-shaped model as described in Figure 1.

Furthermore, T acts in concert with other sex hormones such as estrogens and progesterone in physiological functions. Thus, examining the combined impact of sex hormones, such as T/E2 ratios, may be more informative for understanding the roles of sex hormones in cancer development. Although genetics play an important role in determining sex hormone levels, diet, supplements, and exercise can all tune hormone levels and therefore make interventions possible [225]. Additionally, many agonists and antagonists of AR have been developed and used in cancer treatment, and these methods can potentially be applied to cancers such as bladder, esophagus, head and neck, and stomach.

A major limitation of the existing clinical literature is the lack of consistent adjustment for key confounding variables such as age, BMI, smoking, alcohol intake, and comorbid conditions such as diabetes and metabolic syndrome, which are all known to influence circulating testosterone and AR activity [55,56]. Much more epidemiological work is needed in this area to obtain consistent and convincing results.

In summary, this review focuses on non-reproductive cancers with higher incidence in males, highlighting differences in T levels and potential AR roles in affecting tumorigenesis, progression, treatment, and outcome. Despite mounting evidence of AR’s involvement in these cancers, the precise contribution of T/AR in sex disparities, cancer biology, prognosis, treatment responses, and survival is largely unclear. Closing this knowledge gap may provide novel therapeutic methods, as targeting AR signaling presents a promising yet intricate avenue. To advance our understanding of T/AR’s role and improve AR-targeted therapies, future research should 1) delve into epidemiological and genomic data to characterize T/AR and other sex hormones in a sex-specific manner, thereby guiding mechanistic studies, and 2) identify biomarkers that can stratify patients most likely to benefit from AR-targeted interventions, thus maximizing treatment efficacy.

## Figures and Tables

**Figure 1 cancers-17-03414-f001:**
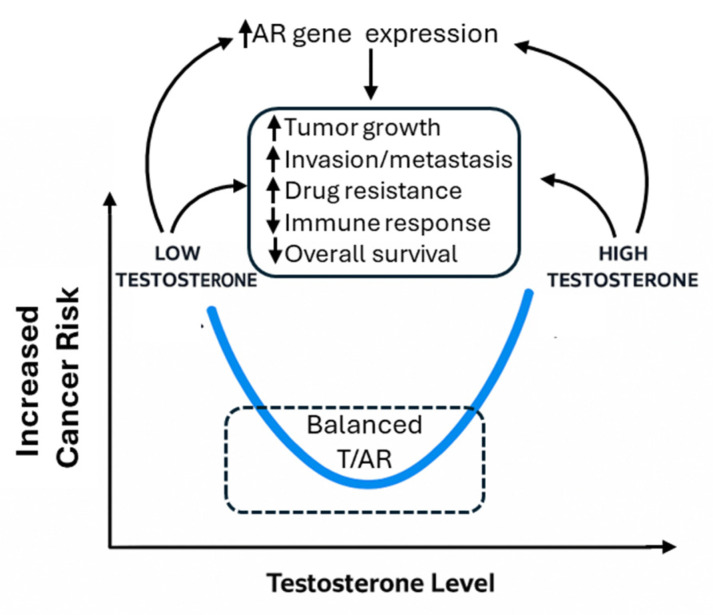
Illustration of U-shaped effects of T/AR axis and their potential pathways.

**Figure 2 cancers-17-03414-f002:**
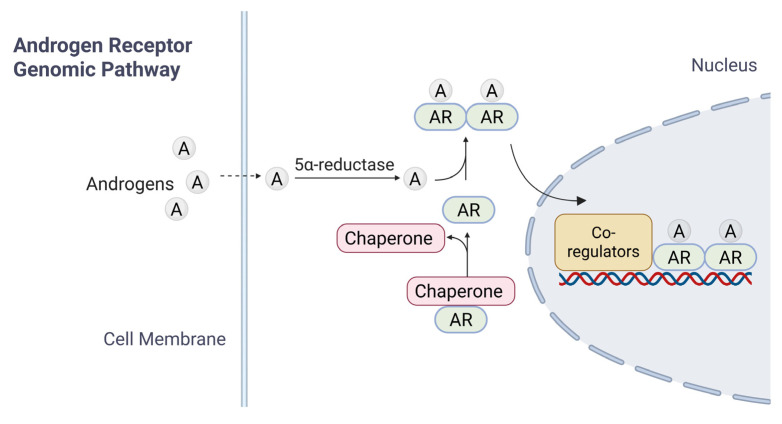
A brief summary of AR signaling. The diagram illustrates the mechanisms of androgen receptor (AR) signaling in response to androgen. In the genomic pathway, androgen diffuses across the cell membrane. Androgen binds to cytoplasmic AR, which is maintained in an inactive conformation by chaperone proteins. Ligand binding induces AR activation, chaperone dissociation, and subsequent nuclear translocation. Within the nucleus, AR-Androgen complexes bind to androgen response elements (AREs) on DNA, recruiting co-regulatory proteins to modulate transcription of target genes.

**Figure 3 cancers-17-03414-f003:**
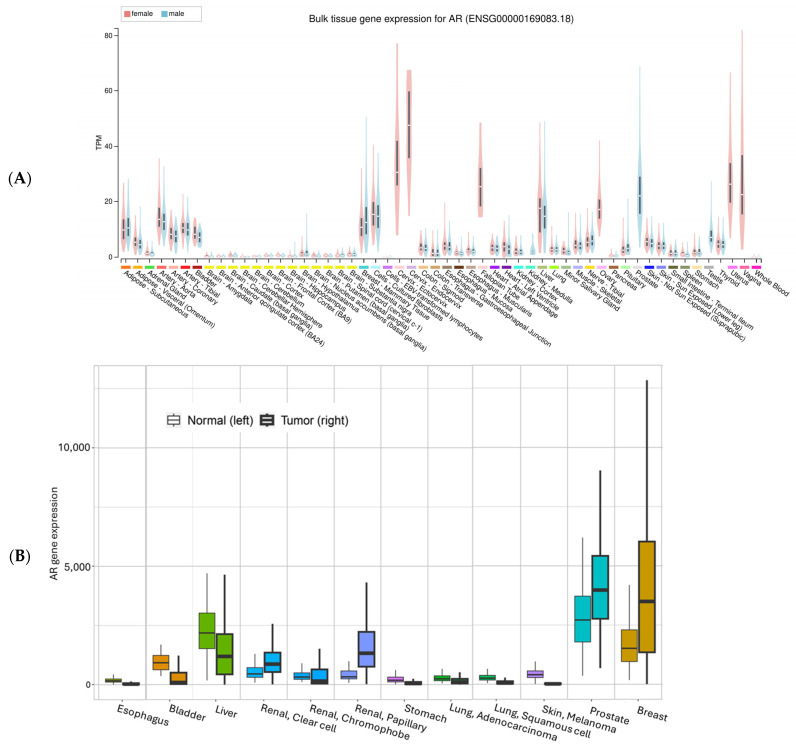
AR mRNA expression in normal (**A**) and normal vs. tumor tissues. The gene expression in normal tissue was based on GTEx data (Genotype-Tissue Expression, with all tissues included), the normal vs. tumor expression was based on an integrated database which included TCGA (The Cancer Genome Atlas), GTEx, TARGET (Therapeutically Applicable Research to Generate Effective Treatments), and NCBI GEO data [33]. The y-axis in (**B**) is batch-adjusted and normalized counts. All comparisons showed significant differences (α < 0.05).

**Figure 4 cancers-17-03414-f004:**
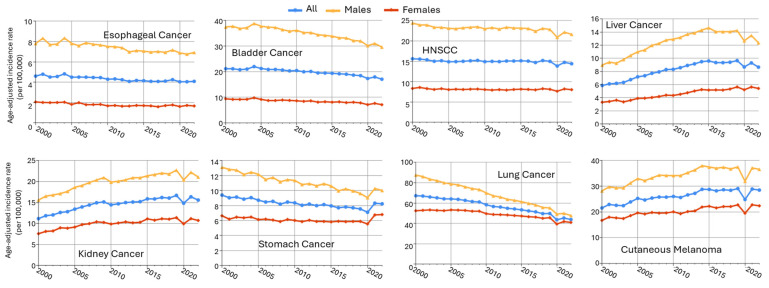
Sex disparities in age-adjusted cancer incidence rates across multiple cancer types (SEER 17 data, 2000–2022). Trends in age-adjusted cancer incidence rates (per 100,000 person-years) from 2000 to 2022 for males (orange), females (red), and all (blue) patients across 8 cancer types. Data were obtained from the SEER Research Limited-Field Database (17 registries; November 2024 submission). Rates were age-adjusted to the 2000 U.S. standard population.

**Table 1 cancers-17-03414-t001:** Male to female incidence ratios (Data: SEER 17 registries, 2000–2022 *).

		Males		Females		All		M/F Ratio	
	SEER Site Code	Rate	95% CI	Rate	95% CI	Rate	95% CI	Ratio	95% CI
Esophageal Cancer	C15	7.37	7.31, 7.42	1.77	1.75, 1.80	4.30	4.27, 4.33	4.16	4.09, 4.23
Bladder Cancer	C67	34.51	34.38, 34.63	8.42	8.36, 8.47	19.66	19.60, 19.73	4.10	4.06, 4.13
Head and Neck Cancer	C00–C14, C32	23.04	22.95, 23.14	8.10	8.05, 8.16	15.00	14.95, 15.05	2.84	2.82, 2.87
Liver Cancer	C22	12.78	12.71, 12.86	4.65	4.61,4.69	8.43	8.39, 8.47	2.75	2.72, 2.78
Kidney Cancer	C64	20.10	20.00, 20.19	10.04	9.98, 10.10	14.66	14.61, 14.72	2.00	1.99, 2.02
Stomach Cancer	C16	10.93	10.86, 11.00	6.09	6.04, 6.14	8.22	8.18, 8.26	1.80	1.78, 1.81
Cutaneous Melanoma	C44	34.62	34.50, 34.75	20.41	20.32, 20.49	26.43	26.36, 26.51	1.70	1.69, 1.71
Lung Cancer	C34	66.06	65.89, 66.23	48.20	48.07, 48.33	55.88	55.77, 55.98	1.37	1.37, 1.38

* Data was obtained with SEER*Stat software (version 9.0.41) and Database: Incidence—SEER Research Data, 17 Registries, Nov 2024 Sub (2000–2022).

**Table 2 cancers-17-03414-t002:** Testosterone and SHBG association with EAC.

	Petrick et al., 2019 [69]	Xie et al., 2020 [70]	Meta-Study (Xie et al., 2020 [70])	Meta-Study (Liu et al., 2023 [71])
	n = 518 (Men Only)	n = 488 (Men Only)		
DHEA	** 0.62 (0.47 to 0.82) **			
DHT	0.94 (0.71 to 1.24)			
T/E2	1.09 (0.79 to 1.51)	**0.46 (0.23, 0.91) ^a^**	0.64 (0.34, 1.22)	0.66 (0.36, 1.21)
Testosterone	0.91 (0.71 to 1.16)	**0.44 (0.22, 0.88) ^b^**	**0.60 (0.38, 0.97)**	0.99 (0.81, 1.20)
		0.56 (0.27, 1.13) ^c^		
SHBG	0.92 (0.66 to 1.28)	0.79 (0.38, 1.64)	0.93 (0.58, 1.49)	1.12 (0.97, 1.30)

^a^, the highest to lowest quartile; ^b^, adjusted for tobacco smoking and physical activity; ^c^, further adjusted for BMI. Bold numbers indicate significant association.

**Table 3 cancers-17-03414-t003:** AR expression and co-regulators/targets.

	Location of AR	Detection Methods	Tumor vs. Normal	Known Co-Regulators or Targets	References
Esophagus Cancer	Nucleus	IHC, WB	upregulated in tumor	UGT2B15, MMP2, pAKT, AP-1, GATA3, IL-6, DUSP4, FOSB	[72,73,74,75]
Bladder Cancer	Nucleus	IHC	upregulated in tumor	CD24, EGFR, VEGF, ncRNAs, UGT1A, FOXO1, GATA3, ADAR2	[76,77,78,79,80,81,82,83,84,85,86,87,88,89,90,91,92,93,94]
HNSCC	Nucleus and cytoplasmic	IHC	up or down, depending on subtypes and stages	VEGF, MMP9, RBAP48, TP53, EGFR, AKT	[95,96,97,98,99,100,101,102,103,104,105]
Liver Cancer	Nucleus	IHC	up or down, depending on stages	EZH2, BIRC7, IGFBP3, NTSR1	[106,107,108,109,110]
Kidney Cancer	Nucleus	IHC	no difference	lncRNA-TANAR/TWIST1, circHIAT1	[111,112,113,114]
Stomach Cancer	Nucleus	qRT-PCR, WB, IHC	upregulated in tumor	LAMA4, YAP1, GSK3b/b-catenin, Snail, TWIST, STAT3	[115,116,117,118,119,120,121,122,123,124,125,126]
Melanoma	Nucleus, cytoplasmic	IF, IHC	unknown	Ku70/Ku80/RNA pol II, ICAM1, STING, FUT4, EGFR, TGF-b	[53,62,127,128,129,130,131,132]
Lung Cancer (NSCLC)	Nucleus, some cytoplasmic	IHC	upregulated in tumor	EGFR, p38, miR-23a-3p, circular-SLCO1B7/miR-139-5p, miR-224-5p	[48,126,133,134,135]

**Table 4 cancers-17-03414-t004:** Effect of androgens on cancer incidence.

	Association	Preliminary Conclusion	References
Esophagus	mixed: DHEA—yes; T or DHT—no and yes; SHBG—no	Higher T may be protective	[69,70,71]
Bladder Cancer	mixed: T—no and yes; MR study—no	Likely no association	[111,146,147]
Head and Neck Cancer	T—No, but HRT seems protective in women	Insufficient data	[100,104]
Liver Cancer	Total T and SHBG—yes in men, no in women;	Higher T may elevate risk	[71,111,148,149]
Kidney Cancer	No association	No association	[110,138]
Stomach	Total or free T—no; SHBG—yes in men, no in women	Likely no association	[71,111]
Melanoma	Total or free T—yes in men, no in older women	Likely higher T is associated with melanoma risk, but it may be confounded by UVR	[111,150,151,152]
Lung Cancer	No association	No association	[111]

**Table 5 cancers-17-03414-t005:** Effects of AR in tumor progression and patient survival.

	Results	Preliminary Conclusion	References
Esophageal Cancer	Higher AR is associated with poorer survival	AR likely exhibits harmful effects on survival	[69,70,71]
Bladder Cancer	Higher AR in cancer than in normal tissue; AR facilitates chemically induced tumors; 5α-reductase inhibitors improve patient survival; castration inhibits tumor growth; absence of AR increases recurrence.	AR likely promotes tumor formation and growth (harmful)	[77,78,79,80,81,82,83,84,85,86,87]
Head and Neck Cancer (c76.0)	Higher AR is associated with cell invasion and poor survival of patients; AR is proposed to be a treatment target.	AR likely exhibits harmful effects on survival	[50,104,156]
Liver Cancer	Mixed results: Higher AR is linked to recurrence and poor survival, as well as better survival, depending on studies.	Controversial	[96,164,165,166,167,168,169,170,171,172]
Kidney Cancer	AR promotes vasculogenesis but is also linked to better patient survival.	Controversial	[173,174,175,176]
Stomach Cancer	AR is linked to drug resistance, likely promotes tumors.	AR likely exhibits harmful effects	[119,120,121,122,123]
Cutaneous Melanoma	AR is linked to cell proliferation, invasiveness, and drug resistance, but higher AR patients exhibit better survival.	Controversial	[40,127,128,129,130,131,132,210,211,212,223]
Lung Cancer	AR collaborates with EGFR to promote tumor growth, but AR-positive patients show better survival.	Controversial	[133,135,199,200,201,208]

## Data Availability

The original datasets are publicly available from the Surveillance, Epidemiology, and End Results (SEER) Program of the National Cancer Institute.

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
