# Peer review of "Testosterone and Androgen Receptor in Cancers with Significant Sex Dimorphism in Incidence Rates and Survival"

_cancers, 2025, doi:10.3390/cancers17213414_

Round 1
Reviewer 1 Report
Comments and Suggestions for Authors
Major revision recommendation:
This comprehensive review on testosterone and AR in sex-dimorphic cancers is timely and well-structured. However, the manuscript would benefit from a more critical synthesis of contradictory findings, especially regarding AR's dual roles. Clarifying the U-shaped testosterone hypothesis and incorporating more recent mechanistic studies would strengthen the conclusions. Additionally, the tables should be better integrated into the text to improve readability and impact. Major revisions are recommended before publication.
- The introduction presents a broad overview but could be strengthened by a more precise, central hypothesis. Please explicitly state the working hypothesis that guided this review (e.g., "We hypothesize that the T/AR axis contributes to sex disparities in non-reproductive cancers not linearly, but in a context- and cancer-type-dependent manner, potentially following a U-shaped model."
- The U-shaped relationship for testosterone is intriguingly introduced only in the conclusion. This is a potentially major conceptual framework that should be introduced much earlier (e.g., in the introduction or Section 2) and its evidence (or lack thereof) should be systematically evaluated for each cancer type throughout the review, not just mentioned at the end.
- The review is largely descriptive, listing associations between T/AR and outcomes. It would be significantly improved by dedicating more space to the mechanisms(e.g., specific downstream signaling pathways like PI3K/AKT, RAS/RAF, JAK/STAT; cross-talk with estrogen receptor signaling; epigenetic modifications induced by AR activation).
- The proposed figures are helpful but could be more informative. Consider adding a schematic that visually represents the proposed U-shaped model for different cancer types. Another figure could map the complex cell-type-specific interactions of AR signaling within the TME.
- Section 2 (Mechanisms): This section is a good start but lacks detail. Please elaborate on the specific AR coregulators (e.g., SRCs, NCOAs) and pioneer factors (e.g., FOXA1, GATA2) that are relevant in the context of non-reproductive cancers. Their differential expression could explain tissue-specific effects.
- Table 1 (AR Expression): This table is useful but should include more information. Please add columns for: 1) The primary cellular localization of AR (nuclear vs. cytoplasmic), which is functionally important, and 2) The methodology used for assessment (IHC, RNA-seq, etc.), as this impacts the interpretation.
- Many clinical studies fail to control for key confounders like age, BMI, smoking status, and concomitant illnesses, which all influence hormone levels. The discussion should explicitly acknowledge this major limitation of the existing literature.
Author Response
Reviewer # 1:
Major revision recommendation:
This comprehensive review on testosterone and AR in sex-dimorphic cancers is timely and well-structured. However, the manuscript would benefit from a more critical synthesis of contradictory findings, especially regarding AR's dual roles. Clarifying the U-shaped testosterone hypothesis and incorporating more recent mechanistic studies would strengthen the conclusions. Additionally, the tables should be better integrated into the text to improve readability and impact. Major revisions are recommended before publication.
Thanks for the constructive suggestions. Tables have been re-arranged.
- The introduction presents a broad overview but could be strengthened by a more precise, central hypothesis. Please explicitly state the working hypothesis that guided this review (e.g., "We hypothesize that the T/AR axis contributes to sex disparities in non-reproductive cancers not linearly, but in a context- and cancer-type-dependent manner, potentially following a U-shaped model." We also cited the tables differently and hence placed where it was first mentioned.
Response: Thanks for this suggestion. We added in the beginning of “Introduction” section about our hypothesis and why this review is written, incorporating the reviewer’s suggested lines. This also includes added Figure 1 for illustration of U-shaped model.
- The U-shaped relationship for testosterone is intriguingly introduced only in the conclusion. This is a potentially major conceptual framework that should be introduced much earlier (e.g., in the introduction or Section 2) and its evidence (or lack thereof) should be systematically evaluated for each cancer type throughout the review, not just mentioned at the end.
Response: We attempted to examine each cancer type for potential U-shape relationship and incorporate the findings. However, there is very limited publications on this aspect. However, we added Figure 1 to illustrate our model.
- The review is largely descriptive, listing associations between T/AR and outcomes. It would be significantly improved by dedicating more space to the mechanisms(e.g., specific downstream signaling pathways like PI3K/AKT, RAS/RAF, JAK/STAT; cross-talk with estrogen receptor signaling; epigenetic modifications induced by AR activation).
Response: Thanks for this important suggestion. We included additional table (Table 3) for AR targets or co-regulators.
- The proposed figures are helpful but could be more informative. Consider adding a schematic that visually represents the proposed U-shaped model for different cancer types. Another figure could map the complex cell-type-specific interactions of AR signaling within the TME.
Response: Thank you for the suggestion. Figure 1 (U-shaped model) is added. We also outlined a few important AR functions related to tumoirgenesis, progression and outcome in Figure 1. The specific publications of AR function in TME can be found in PMDI: 38863718, hence we only referred to this reference in our manuscript without re-drawing the illustration. One other main reason is that there is no sufficient research in cell-type specific AR functions. If we draw such a figure, it would be as general as the rest of publications regarding AR function.
- Section 2 (Mechanisms): This section is a good start but lacks detail. Please elaborate on the specific AR coregulators (e.g., SRCs, NCOAs) and pioneer factors (e.g., FOXA1, GATA2) that are relevant in the context of non-reproductive cancers. Their differential expression could explain tissue-specific effects.
Thank you for the suggestion. We now listed as many co-regulators and targets as we can found in the literature in Table 3. Additionally we added more details in each cancer type regarding mechanisms.
- Table 1 (AR Expression): This table is useful but should include more information. Please add columns for: 1) The primary cellular localization of AR (nuclear vs. cytoplasmic), which is functionally important, and 2) The methodology used for assessment (IHC, RNA-seq, etc.), as this impacts the interpretation.
Response: Thanks for the suggestion. We inserted a new table (current Table 3) to summarize AR expression, location, methods of detection and co-regulators/targets as we feel it is a bit awkward to include this information in Table 1 which is about incidence ratios.
Additionally, we added a figure to show AR expression in various normal tissues and normal vs tumor tissues (Figure 3).
- Many clinical studies fail to control for key confounders like age, BMI, smoking status, and concomitant illnesses, which all influence hormone levels. The discussion should explicitly acknowledge this major limitation of the existing literature.
Response: We acknowledged this in our discussion now.
Reviewer 2 Report
Comments and Suggestions for Authors
This review examines sex differences in cancer incidence and survival, with a focus on the role of testosterone and androgen receptor (AR) signaling. Epidemiological and molecular evidence suggests that elevated testosterone levels may increase cancer risk in certain types of cancer, although AR activity does not always align with testosterone levels in tumor development and progression. AR appears to promote tumor growth and poor survival in several cancers, including esophagus, bladder, head and neck, and stomach cancers. The authors highlight the need for further research to clarify a potential U-shaped relationship between testosterone and cancer risk and to explore AR-based therapeutic opportunities. Overall, the authors made a good effort in summarizing the current literature on the topic which is significant in the field of steroid hormone signaling in cancer. The authors also treated the different sub-topics with some details. The figures and tables are generally informative. This reviewer has a few comments intended for improvement of its clarity and significance.
- The review highlights that androgen levels contribute to sexual dimorphism in cancer. The authors should also summarize and/or discuss the relationship between the effects of systemic hormone levels versus tumor-intrinsic AR activity.
- In 2.1, could the authors expand on AR expression and functions in skin and mucosal tissues, which are relevant to several cancers (HNSCC, melanoma) with sex dimorphism discussed?
- The manuscript would benefit from an overview of AR signaling in cancer, including its roles in growth, EMT, metastasis, and therapeutic responses, to provide sufficient context for a general audience.
- Are there reported mutations of AR signaling components in non-prostate cancers? If yes, the authors should discuss them in this review.
- The authors mainly discuss the level of testosterone in cancer risk, how about the level of dihydrotestosterone (DHT)?
- HPV-related and non-HPV-related HNSCC have distinct prognoses. Could the authors discuss whether sex dimorphism and AR signaling differ between these two subtypes?
- Since tumorigenesis and prevention are discussed in the manuscript, the authors should include the studies about testosterone and AR in premalignant lesions of HNSCC, such as oral leukoplakia.
- In prostate cancer, resistance to AR-directed therapies is a major challenge. Do the authors anticipate similar resistance mechanisms, such as mutations, splice variants, or bypass pathways, arising in non-prostatic cancers? If yes, what strategies could potentially address them?

Author Response
Reviewer #2:
Comments and Suggestions for Authors
This review examines sex differences in cancer incidence and survival, with a focus on the role of testosterone and androgen receptor (AR) signaling. Epidemiological and molecular evidence suggests that elevated testosterone levels may increase cancer risk in certain types of cancer, although AR activity does not always align with testosterone levels in tumor development and progression. AR appears to promote tumor growth and poor survival in several cancers, including esophagus, bladder, head and neck, and stomach cancers. The authors highlight the need for further research to clarify a potential U-shaped relationship between testosterone and cancer risk and to explore AR-based therapeutic opportunities. Overall, the authors made a good effort in summarizing the current literature on the topic which is significant in the field of steroid hormone signaling in cancer. The authors also treated the different sub-topics with some details. The figures and tables are generally informative. This reviewer has a few comments intended for improvement of its clarity and significance.
- The review highlights that androgen levels contribute to sexual dimorphism in cancer. The authors should also summarize and/or discuss the relationship between the effects of systemic hormone levels versus tumor-intrinsic AR activity.
Respond: We added a paragraph in Section 2.5 (Current knowledge about Androgens and AR in diseases
- In 2.1, could the authors expand on AR expression and functions in skin and mucosal tissues, which are relevant to several cancers (HNSCC, melanoma) with sex dimorphism discussed?
Respond: We added Figure 3 showing AR mRNA expression in normal and tumor tissues from large genomics databases. Esophagus mucosa and skin data are included. For AR function in HNSCC and skin, we incorporated additional information in each cancer type under sections of 3.3 and 3.8.
- The manuscript would benefit from an overview of AR signaling in cancer, including its roles in growth, EMT, metastasis, and therapeutic responses, to provide sufficient context for a general audience.
Response: Section 2.5 is added to address this point.
- Are there reported mutations of AR signaling components in non-prostate cancers? If yes, the authors should discuss them in this review.
We are not able to find any more in the pubmed.
- The authors mainly discuss the level of testosterone in cancer risk, how about the level of dihydrotestosterone (DHT)?
This is a good question. DHT is a potent activator for AR but its circulating level is only 1/10 of testosterone. Additionally, DHT seems to exert more localized effects such as male genital development and prostate function, and hence are mentioned only when data is available (we did mentioned DHT levels in bladder cancer, Section 3.2, and in Table 2 and Table 4.
- HPV-related and non-HPV-related HNSCC have distinct prognoses. Could the authors discuss whether sex dimorphism and AR signaling differ between these two subtypes?
We added a paragraph in Section 3.4 to include this information, thanks.
- Since tumorigenesis and prevention are discussed in the manuscript, the authors should include the studies about testosterone and AR in premalignant lesions of HNSCC, such as oral leukoplakia.
We couldn’t find well-established studies linking T and AR expression in premalignant lesions.
- In prostate cancer, resistance to AR-directed therapies is a major challenge. Do the authors anticipate similar resistance mechanisms, such as mutations, splice variants, or bypass pathways, arising in non-prostatic cancers? If yes, what strategies could potentially address them?
Direct AR mutations are rarely reported outside the prostate. In non-prostatic cancers, clinical data are limited.
Reviewer 3 Report
Comments and Suggestions for Authors
Lin et al. conducted a comprehensive narrative review that synthesizes epidemiologic and molecular evidence on testosterone and the androgen receptor in non-reproductive cancers that exhibit male-female disparities in incidence and/or outcomes. While the review addresses a significant topic and encompasses a wide range of literature, the authors have not covered numerous aspects, which warrants some considerations before publication.
1. Abstract: Please add 1–2 concrete, balanced exemplars (e.g., “higher AR expression associates with poorer survival in ESCC, whereas AR–survival associations in HCC and melanoma are inconsistent”).
2. I have a concern as I have not seen a method section as it is required for a review. Please provide search databases (e.g., PubMed/Embase/Web of Science), search dates (through YYYY-MM), core strings (androgen/NR3C4/testosterone + site names), inclusion/exclusion (English, humans/animals, study designs), and a brief quality appraisal approach (e.g., ROBINS-I for observational studies, RoB2 for RCTs, SYRCLE for animal studies). Also, could you please clarify how MR studies and UK Biobank analyses were prioritized, and whether preprints were included?
3. Regarding the epidemiology/SEER analyses (Figure 2 & Table 1), I believe that you should re-run and document SEER queries with correct site groups for each cancer (e.g., head & neck: C00–C14 and C32; esophagus: C15; bladder: C67; stomach: C16; liver: C22; kidney: C64; melanoma: C43; lung/bronchus: C34). Authors should provide exact parameters (SEER database version; incidence years; behavior=malignant; age-adjustment standard; race/ethnicity aggregation; multiple primaries rules). Also, you should present rates with 95% CI and M/F ratios with CI; fix duplicated CI columns; ensure Figure 2 labels and values match Table 1 and text.
4. Mechanisms of AR signaling: Please cite the figure as adapted or original and ensure permission where needed and briefly distinguish genomic vs non-genomic actions (you appropriately downplay the latter) and standardize gene/protein formatting (gene NR3C4 italicized, protein AR in Roman).
5. For each cancer site, authors should split “Epidemiology/Exposure” and “Tumor Biology/AR data”, and add evidence tables summarizing the population, design, exposure (total T, free T, SHBG; AR IHC or mRNA), outcomes, effect sizes (HR/OR with CI), and key confounders.
- Esophagus: Please add a table listing two prospective studies that found protective associations for T or T/E2 in EAC and the meta-analysis showing SHBG risk—clearly label inconclusive overall evidence.
- Bladder: Please discuss sex-specific prognosis disadvantage in women separately from incidence and caution against inferring causality.
- Head & Neck: Please clarify that no robust prospective link between T and HNC has been shown; note HRT pooled analysis in women and separate HPV-positive OPSCC where hormone biology may differ and finally summarize AR distribution and the limited clinical data on AR-targeting.
- Liver: Please present conflicting data carefully [(e.g., elevated T and SHBG correlate with risk, yet tumor AR shows dual roles)-tumorigenesis vs metastasis] and I suggest adding a schematic contrasting these phase-specific roles.
- Kidney: Please emphasize the contradictory nature of AR associations (vasculogenesis and invasion vs better survival), and that Biobank studies show no T–risk link.
- Stomach: Authors should make clear that T is not associated, whereas SHBG may be in men; AR appears to contribute to chemoresistance (e.g., LAMA4, YAP1 pathway). Also, they should quantify study sizes and endpoints where possible.
- Lung: I believe that you should keep causal claims modest; AR can collaborate with EGFR in models, yet clinical signals are mixed (some analyses report better survival with AR positivity). Also, please distinguish histologies (LUAD vs LUSC vs SCLC).
- Melanoma: Please integrate TCGA and experimental data indicating AR-driven invasiveness alongside reports of better survival with higher AR—explicitly flag discordance between risk and prognosis signals and motivate future work (e.g., non-linear T effects, therapy-induced AR upregulation).
6. Discussion: It’d be better that authors strengthen the synthesis by separating risk (hormone exposure) from tumor behavior (AR in tumor), presenting the U-shape concept as hypothesis-generating, with examples beyond prostate cancer only if supported and outlining testable predictions (e.g., interaction of AR-targeted agents with BRAF/MEK therapy in melanoma; sex-specific biomarker panels combining T, SHBG, T/E2 ratio, and AR IHC).
7. Substantive English editing is needed to correct typos.
Author Response
Reviewer 3
Comments and Suggestions for Authors
Lin et al. conducted a comprehensive narrative review that synthesizes epidemiologic and molecular evidence on testosterone and the androgen receptor in non-reproductive cancers that exhibit male-female disparities in incidence and/or outcomes. While the review addresses a significant topic and encompasses a wide range of literature, the authors have not covered numerous aspects, which warrants some considerations before publication.
- Abstract: Please add 1–2 concrete, balanced exemplars (e.g., “higher AR expression associates with poorer survival in ESCC, whereas AR–survival associations in HCC and melanoma are inconsistent”).
Thanks for this suggestion. The sentence is added in the abstract.
- I have a concern as I have not seen a method section as it is required for a review. Please provide search databases (e.g., PubMed/Embase/Web of Science), search dates (through YYYY-MM), core strings (androgen/NR3C4/testosterone + site names), inclusion/exclusion (English, humans/animals, study designs), and a brief quality appraisal approach (e.g., ROBINS-I for observational studies, RoB2 for RCTs, SYRCLE for animal studies). Also, could you please clarify how MR studies and UK Biobank analyses were prioritized, and whether preprints were included?
There are a few types of reviews. As the review sees here, this review is a narrative review, which is not like a systematic review or a meta-analysis that requires literature search for specific terms or period of time.
- Regarding the epidemiology/SEER analyses (Figure 2 & Table 1), I believe that you should re-run and document SEER queries with correct site groups for each cancer (e.g., head & neck: C00–C14 and C32; esophagus: C15; bladder: C67; stomach: C16; liver: C22; kidney: C64; melanoma: C43; lung/bronchus: C34). Authors should provide exact parameters (SEER database version; incidence years; behavior=malignant; age-adjustment standard; race/ethnicity aggregation; multiple primaries rules). Also, you should present rates with 95% CI and M/F ratios with CI; fix duplicated CI columns; ensure Figure 2 labels and values match Table 1 and text.
The table and figure are redone, thanks for catching this mistake!
- Mechanisms of AR signaling: Please cite the figure as adapted or original and ensure permission where needed and briefly distinguish genomic vs non-genomic actions (you appropriately downplay the latter) and standardize gene/protein formatting (gene NR3C4 italicized, protein AR in Roman).
This figure is original.
- For each cancer site, authors should split “Epidemiology/Exposure” and “Tumor Biology/AR data”, and add evidence tables summarizing the population, design, exposure (total T, free T, SHBG; AR IHC or mRNA), outcomes, effect sizes (HR/OR with CI), and key confounders.
The author arranged the text in this manner and we feel there is not sufficient data to put into a table for all cancer types. However, we added Table 2 to summarize data in esophageal cancer.
- Esophagus: Please add a table listing two prospective studies that found protective associations for T or T/E2 in EAC and the meta-analysis showing SHBG risk—clearly label inconclusive overall evidence.
We inserted Table 2 to summarize these results.
- Bladder: Please discuss sex-specific prognosis disadvantage in women separately from incidence and caution against inferring causality.
Inserted a paragraph to describe possible causes
- Head & Neck: Please clarify that no robust prospective link between T and HNC has been shown; note HRT pooled analysis in women and separate HPV-positive OPSCC where hormone biology may differ and finally summarize AR distribution and the limited clinical data on AR-targeting.
We had the sentence in Section 3.3 to clarify no robust prospective link between T and HNC. We also add OPSCC observation with AR distribution in that Section.
- Liver: Please present conflicting data carefully [(e.g., elevated T and SHBG correlate with risk, yet tumor AR shows dual roles)-tumorigenesis vs metastasis] and I suggest adding a schematic contrasting these phase-specific roles.
This is indeed somewhat confusing – we now included a paragraph to describe circulating hormone levels and tumor AR activities to clarify this issue (Section 2.5, paragraph 4).
- Kidney: Please emphasize the contradictory nature of AR associations (vasculogenesis and invasion vs better survival), and that Biobank studies show no T–risk link.
We added one sentence in the end of the paragraph to re-iterate the contradiction.
- Stomach: Authors should make clear that T is not associated, whereas SHBG may be in men; AR appears to contribute to chemoresistance (e.g., LAMA4, YAP1 pathway). Also, they should quantify study sizes and endpoints where possible.
Slightly modified the Section 3.6 to emphasize these points.
- Lung: I believe that you should keep causal claims modest; AR can collaborate with EGFR in models, yet clinical signals are mixed (some analyses report better survival with AR positivity). Also, please distinguish histologies (LUAD vs LUSC vs SCLC).
“causal” is deleted. I am not sure what the reviewer wants me to do by “distinguish histologies”?
- Melanoma: Please integrate TCGA and experimental data indicating AR-driven invasiveness alongside reports of better survival with higher AR—explicitly flag discordance between risk and prognosis signals and motivate future work (e.g., non-linear T effects, therapy-induced AR upregulation).
Indeed, melanoma may represent another good model for U-shaped relationship of T/AR activities, we actually have new data to support this and our manuscript is under preparation. It is exciting!
- Discussion: It’d be better that authors strengthen the synthesis by separating risk (hormone exposure) from tumor behavior (AR in tumor), presenting the U-shape concept as hypothesis-generating, with examples beyond prostate cancer only if supported and outlining testable predictions (e.g., interaction of AR-targeted agents with BRAF/MEK therapy in melanoma; sex-specific biomarker panels combining T, SHBG, T/E2 ratio, and AR IHC).
We have now included a figure to show U-shaped model and possible pathways involved. However, we also stated that substantial evidence is needed for this model beyond prostate cancer. We added in discussion that the risk of hormone exposure and tumor behavior should be considered separately, along with a paragraph in Section 2.5 to discuss circulating hormone levels with tumor AR activities.
- Substantive English editing is needed to correct typos.
We proof-read the manuscript and corrected as many errors that we could catch, thanks.
Round 2
Reviewer 1 Report
Comments and Suggestions for Authors
Accept in present form
Author Response
Thank you!
Reviewer 3 Report
Comments and Suggestions for Authors
All comments have been addressed well and I appreciate the thoughtful revisions. Just to clarify one earlier point: when I suggested distinguishing between histologies (LUAD, LUSC, SCLC), I meant that these are fundamentally different diseases from a biological and clinical perspective. They vary in terms of cell of origin, common driver mutations, risk factors, prognosis, and in some cases, hormone receptor biology. Because of these differences, any claims related to testosterone or AR signaling really should be analyzed and interpreted separately by subtype rather than grouped under a broad label like “lung cancer” or even “NSCLC.” One way you might frame it in the manuscript is: “Given the distinct biology of LUAD, LUSC, and SCLC, we analyzed androgen-related findings by histologic subtype; associations observed in LUAD were not assumed to generalize to LUSC or SCLC.” This would make the reasoning behind your analysis more transparent and biologically grounded.
Author Response
Thank you very much for clarification. I have added a sentence to make a note of this difference in Section 3.7